# A Review of the Effect of Nano-Silica on the Mechanical and Durability Properties of Cementitious Composites

Haneen AlTawaiha [1], Fadi Alhomaidat [2,*] and Tamer Eljufout [3]

[1] Civil Engineering Department, Faculty of Ma'an College, Al-Balqa' Applied University, Ma'an 71110, Jordan
[2] Civil Engineering Department, Faculty of Engineering, Al-Hussein Bin Talal University, Ma'an 71111, Jordan
[3] Civil Engineering Department, College of Engineering, The University of Texas at Arlington, 701 S. Nedderman Drive, Arlington, TX 76019, USA
* Correspondence: fadi.alhomaidat@ahu.edu.jo

**Abstract:** The incorporation of nanotechnology has led to significant strides in the concrete industry, ushering in innovative construction methodologies. Various nanomaterials, including nano-silica (NS), have undergone comprehensive scrutiny as potential partial substitutes for cement in concrete formulations. This article aims to provide a comprehensive overview of the impacts of NS on several mechanical properties of concrete, encompassing compressive, split tensile, and flexural strengths. Additionally, the review delves into the influence of NS on the concrete's durability, including microstructural characterization and the eradication of structural micropores. NS has demonstrated the capacity to bolster both strength and durability while concurrently diminishing structural micropores. Moreover, this review explores the contemporary status of NS application in cement concrete and presents avenues for prospective research. The assessment of engineering attributes becomes imperative for concrete infused with nano-silica. This encompasses aspects like bond strength, creep, shrinkage, and more. A rigorous evaluation of fresh and hardened properties is necessary to discern the material's thermal and acoustical characteristics. Such a comprehensive understanding contributes to a holistic evaluation of the material's adaptability across diverse applications.

**Keywords:** nano-silica modified concrete; durability; mechanical properties

## 1. Introduction

Concrete has undergone significant changes since its inception, starting with normal concrete. In the 1900s, this concrete form was often utilized for construction projects since it offered adequate strength for all uses [1]. Normal concrete contained fewer than 380 kg/m$^3$ of cement, typical aggregates, moderate water requirements, and a small amount of superplasticizers [2]. However, the emergence of unique structural designs in the 1960s required concrete with a high load-bearing capacity, exceeding 50 MPa up to 95 MPa [3]. High Strength Concrete (HSC) was developed to meet this demand, which can bear loads ranging from 50 MPa to 90 MPa [4]. More cement, more aggregate, less water, and suitable superplasticizers are needed for HSC. Various additives and extra materials were introduced to do this, including nano-silica, fly ash, metakaolin, and other pozzolanic minerals [5,6].

Employing nanoparticles to improve the mechanical behavior of cementitious composites was originally explored in the late 1980s, and research in this field has been ongoing for almost two decades [7]. Due to their remarkable qualities and capabilities, nanomaterials can improve the behavior of concrete [8–10]. Nano-SiO$_2$ is one of the most widely used nanomaterials, made up of pozzolanic materials that can react with cement hydration products [11]. Incorporating Nano-SiO$_2$ improved cement-based performance, such as compressive, flexural strength, water penetration resistance, sulfate attack, and reduced calcium leaching [11–13].

Nanotechnology has experienced significant progress, leading to noteworthy findings concerning particles smaller than 100 nm [14]. These minute particles can augment the mechanical characteristics of diverse materials, including polymers [15] and concrete [16]. Additionally, they hold relevance in various sectors, such as engineering, food, and medicine [17]. Therefore, researchers have concentrated on examining the effects of nano-silica in concrete [18]. Several nanoparticles have been studied, including nano titanium dioxide, nano aluminum oxide, nano iron oxide, nano zinc oxide, and nano-silica [18,19].

According to studies, adding nano-silica to concrete can greatly improve the compressive strength of the material [20]. Additionally, it has been demonstrated that nano-silica can shorten the initial and final setting while accelerating the concrete's early-age strength. This is clarified via the point that nano-silica has a big specific surface area and serves as a solid binder between cement and aggregate [21,22]. Additionally, due to its extremely small particle size, nano-silica exhibits excellent pozzolanic activity [23,24], enabling it to completely fill in all the pores and voids in concrete, including the Interfacial Transition Zone (ITZ), thereby enhancing its strength [20,25,26].

Furthermore, introducing nano-silica enhances calcium silicate hydrate (C-S-H) gel formation, a critical component for concrete strength. This increased concrete hydration, as studied by Norhasri et al. (2017) [27], Barbhuiya et al. (2020) [16], and Mohammed et al. (2018) [28], leads to a higher C-S-H gel content. Simultaneously, the reaction between nano-silica and portlandite-$Ca(OH)_2$ reduces the concrete's portlandite content, resulting in a denser concrete structure.

Despite numerous studies on the impact of nano-silica on concrete, there is a lack of comprehensive research that covers all its effects in a single document. This study aims to bridge that gap by thoroughly investigating the concrete's mechanical characteristics, durability, and microstructural characteristics containing nano-silica. The analysis is based on a meticulous review of approximately one hundred research papers demonstrating nano-silica's diverse impacts on concrete.

## 2. Nanomaterials

Nanomaterials are materials that have been decreased in component to a range of 1–100 nm or contain at least a single dimension inside this nanoscale range in a three-dimensional space. Nanostructured materials and nanostructured components are the two main subcategories of nanomaterials. While nanostructured components have at least one structural component with an outside dimension inside the nanometer range, nanostructured materials are distinguished by having structural dimensions that are in the nanoscale range. This categorization is based on the external dimensions of the structural elements of the materials. [29,30]. Nanomaterials possess unique characteristics that set them apart from conventional materials. For instance, cement mortar incorporating nano-silica or nano-$Fe_2O_3$ demonstrated increased strength in compression and flexure after 28 days of measurement compared to the blank group, indicating an improved performance due to the inclusion of nanomaterials [31]. Similarly, nano-$Al_2O_3$ ceramics exhibited higher flexural strength than micro-scale monolithic alumina ceramics, highlighting the enhanced mechanical properties of nanomaterials [32]. Table 1 presents the types of nanomaterial particles documented in the literature.

**Table 1.** The previous studies examined the utilization of various nanomaterials and their corresponding substitution ratios.

| Reference | Type of Nanomaterial | Type of Concrete | Type of Use | Remarks |
|---|---|---|---|---|
| Amin & Abu el-Hassan (2015) [33] | (Ni ferrite) and (Cu-Zn ferrite) were utilized in the experiment together with 15 nm nano-silica. | High-strength concrete | In the investigation, weight doses of 1%, 2%, 3%, 4%, and 5% of nano-sillica, Ni, and Cu-Zn ferrite were added to cementitious materials. | Comparing samples of concrete with nano-ferrite to samples of concrete with nano-silica, the latter produced compressive strength that were superior by an estimated 10%. |
| Ren et al. (2018) [20] | The experiment employed nano-titanium dioxide particles with a diameter of 10 nm and nano-silica particles with a diameter of 20 nm. | Normal concrete | In the study, cement was substituted with nano-silica and nano-$TiO_2$ to varying degrees (1%, 3%, and 5%, respectively). | With a mass concentration of 3%, NS and NT may each maximally increase the compressive strength of concrete by 16% and 9%, respectively. |
| Zhao et al. (2012) [34] | The nano-silica particle dimension averages employed in the investigation was about 100 nm. | Normal concrete | In the study, nano-silica was utilized at different weight percentages, including 0%, 5%, 15%, and 20%. | The ability of compression and frost resistance increases by 20% when nano-$SiO_2$ concentration is 10% compared to conventional concrete. |
| Shaikh & Supit (2014) [35] | In the experiment, nano-$CaCO_3$ powder (40 to 50 nm) was employed. | Fly ash concrete | The study incorporated Nano-$CaCO_3$ into cement at weight dosages of 1%, 2%, 3%, and 4%. | Results demonstrate that among all nano-$CaCO_3$ concentrations, 1% $CaCO_3$ nanoparticles had the maximum compressive strength, which was also 22% greater than that of cement mortar. |
| Chithra et al. (2016) [36] | The solution under consideration is a colloidal dispersion of nanoparticles in water, which has a density range of 1.3 to 1.32. | High-performance concrete | The study conducted with nano-silica involved replacing different percentages of cement by weight, specifically 0%, 0.5%, 1%, 1.5%, 2%, 2.5%, and 3%. | The addition of nano-silica to cement mortars that used 40% copper slag as a substitute for fine aggregate enhanced the compressive strength by 2%. |
| Salemi & Behfarnia (2013) [37] | Nanoparticles of 20 nm diameter silicon and 8 nm diameter aluminum oxide were the materials employed in the investigation. | Concrete pavement | In the investigation, NS at 3%, 5%, and 7% and nano-$Al_2O_3$ at 1%, 2%, and 3% were used to substitute cement to varying degrees by weight. | According to experimental findings, adding 5% nano-silica to cementitious materials increases concrete's compressive strength and frost resistance by up to 30% and 83%, respectively. |
| Mohamed (2016) [38] | Nano-silica and nano-clay (NC) | Normal concrete | The study involved substituting cement at varying weight percentages, ranging from 0.5% to 10%. | Nano-silica and nano-clay both significantly increase the compressive strength of high-performance concrete by 18% and 11%, respectively. |

**Table 1.** *Cont.*

| Reference | Type of Nanomaterial | Type of Concrete | Type of Use | Remarks |
|---|---|---|---|---|
| Wu et al. (2016) [39] | Nano-CaCO$_3$ elements and nano-silica particles with diameters ranging from 5 to 35 nm and 15 to 105 nm, respectively, were used in the study. | High strength concrete | In the experiment, paste was substituted by mass with different percentages of nano-CaCO$_3$, specifically 0%, 1.6%, 3.2%, 4.8%, and 6.4%, as well as with nano-silica at 0%, 0.5%, 1.0%, 1.5%, and 2.0% of the mass of cement. | While nano-SiO$_2$ UHSC combinations exhibited a continuous and strong rise in strength with age up to 7 days, nano-CaCO$_3$ UHSC mixtures essentially showed constant strength between 3 and 7 days, but a fast increase beyond that |
| Li et al. (2015) [40] | The study used both nano-silica nanoparticles (20 nm) and nano-limestone nanoparticles (15–80 nm). | Ultra-high-performance concrete | The experiment involved partially replacing cement by mass with nano-silica at 0.5%, 1.0%, 1.5%, and 2.0%, as well as with nano-limestone at 1.0%, 2.0%, 3.0%, and 4.0%. | The increase in the flexural to compressive strengths ratio of 1.0% NS-integrated UHPC matrix with W/B ratios of 0.16 is 36%. |
| Gao et al. (2017) [41] | Nano-silica nanoparticles with an average particle size of 15 nm were used in the study, as were nano-sillica nanoparticles with a medium grain size of 50 nm. | Road flyash concrete | At quantities of 3%, 2%, and 1% of the composition of cementitious materials, silica fume and nano-silica were both used in the experiment. | Compared to the reference concrete, the concrete containing 2% NS at 28 days saw a 124.8% increase in drying shrinkage. |
| Torabian et al. (2016) [42] | The material used in the study was composed of nano-silica nanoparticles, which had an average particle size of 20 nm. | Normal concrete | The study involved using nano-silica to replace cement in different quantities, specifically 0.5%, 1%, and 1.5%. | A 41% increase in strength is achieved by adding 1.5% NS to concrete with a w/b ratio of 0.65. |
| Said et al. (2012) [43] | The substance utilized in the study consisted of nano-silica nanoparticles that had a medium grain size of 35 nm. | Normal concrete | During the experiment, various quantities of nano-silica nanoparticles, especially 6% and 12% by weight, were introduced to the cementitious materials. | With the addition of nano-silica, the strength increased up to 6% at all curing ages. |
| Hosseini et al. (2017) [44] | The experiment utilized nano-clay elements that had a density of 1660 kg/m$^3$. | Self-compacting concrete | The researchers substituted cement with varying proportions of nano-clay, which included 0.25%, 0.5%, 0.75%, and 1% of the total weight of the cement. | At 56 days, the addition of 0.25 and 50% nano-clay increased compressive strength by 15% and 14%, respectively. |

Nanoparticles possess unique mechanical properties that arise from their volume, surface, and quantum effects. When added to a material, they result in a smaller grain size, resulting in the formation of an intragranular or intergranular structure. This, in turn, improves the quality of the grain border as well as enhances the physical characteristics of the material [45]. When nanoparticles are added, they dramatically increase the mechanical

characteristics of numerous materials. Illustratively, the augmentation of mechanical properties in materials via nanoparticles is evident in the work of Saba et al. (2016) [46]. Their study demonstrated a notable enhancement in mechanical behavior by incorporating a 3% nanoscale oil palm hollow fruit fiber filler into kenaf epoxy alloys.

A significant stride has been made in understanding the mechanical characteristics of metal nanomaterials. Table 2 presents detailed information on the physical characteristics of nanomaterials.

**Table 2.** Mechanical properties of metal nanomaterials [47,48].

| Sample | Vickers Hardness Gpa | Fracture Toughness (MPa Öm) | Fracture Strength (MPa) | Ultimate Tensile Strength (MPa) | Impact Strength (J/cm$^2$) |
|---|---|---|---|---|---|
| Monolithic $Al_2O_3$ | 17.8 | $3.6 \pm 0.3$ | $536 \pm 35$ | - | - |
| $Al_2O_3$/Cu(oxide) | 17.0 | $4.9 \pm 0.7$ | $819 \pm 53$ | - | - |
| $Al_2O_3$/Cu(nitrate) | 17.2 | $4.8 \pm 0.2$ | $953 \pm 59$ | - | - |
| $Al_2O_3$/Ni-Co | 19.0 | $4.3 \pm 0.5$ | $1070 \pm 72$ | - | - |
| AA6061/nano-SiC | 96.4 | - | - | 190.2 | 15.5 |
| AA6061/nano-$B_4C$ | 69.4 | - | - | 201.5 | 12.4 |
| AA6061/1.5SiC + 1.5$B_4C$ | 173.9 | - | - | 280.2 | 19.7 |

The higher performance of nanocomposites containing metal nanoparticles over monolithic $Al_2O_3$ is proof that, according to [47], the addition of metal nanoparticles to nanomaterials increases their fracture toughness and strength. Metal particle pinning inhibits the formation of the $Al_2O_3$ matrix, resulting in a lower grain size and grain refining, which eventually improves the mechanical characteristics. In nanocomposites containing nano-Cu, the resulting hardness is less than that of $Al_2O_3$ because of the lower stiffness of Cu compared to $Al_2O_3$. However, in contrast, nanocomposites with nano-Ni-Co show an upper hardness than $Al_2O_3$ because of the greater toughness of Ni-Co relative to $Al_2O_3$. The final three data sets in Table 2 show that hybrid materials outperform single-reinforced composites in maximum toughness, impact resistance, and final tensile strength, which is most likely because of the reaction of SiC and $B_4C$. These findings underscore the impact of metal nanoparticles on the structural behavior of nanomaterials.

Table 3 presents the physical properties of nonmetallic nanomaterials. The mechanical properties of skutterudites decrease with the inclusion of carbon nanotubes in the resulting nanocomposite. It is hypothesized that the reduction in the physical characteristics of the resulting nanocomposite is produced via the construction of agglomerates of nanotubes made of carbon within the skutterudites. These agglomerates can operate as planes of slip that promote fracture propagation, lowering the nanocomposite's overall mechanical performance. As a result, sample fracturing occurs even at modest mechanical stresses, as described by Schmitz et al. (2017) [49]. Contrarily, most organic nanostructures are flexible and lack mechanical characteristics like hardness and compressive strength. The last five entries in Table 3 show a decrease in tensile strength as the concentration of nano-HA increases, which may be due to a poor boundary between nano-HA and nano-PLLA. With the addition of nano-HA, the flexural strength of the nanocomposites first rises, then falls beyond a certain threshold. The maximum bending strength of 156.8 MPa is attained at a nano-HA content of 20%.

**Table 3.** Mechanical characteristics of nonmetallic nanomaterials [46,49,50].

| Sample | Vickers Hardness Gpa | Compressive Strength (MPa) | Flexural Strength (MPa) | Tensile Strength (MPa) | Elongation at Break (%) | Young's/Bending Modulus (GPa) |
|---|---|---|---|---|---|---|
| p-type skutterudites | 576 ± 52 | 630 ± 20 | 105 ± 10 | - | - | 44 ± 8 |
| p-type skutterudites + 0.5 wt% multi-walled carbon nanotubes | 513 ± 52 | 355 ± 15 | 65 ± 70 | - | - | 40 ± 6 |
| p-type skutterudites + 1.0 wt% multi-walled carbon nanotubes | 563 ± 85 | 320 ± 15 | 54 ± 7 | - | - | 39 ± 8 |
| p-type skutterudites + 1.5 wt% multi-walled carbon nanotubes | 569 ± 70 | 255 ± 10 | 45 ± 5 | - | - | 33 ± 10 |
| oil palm empty fruit bunch fiber | - | - | - | 50–400 | 8.0–18.0 | 1.0–9.0 |
| Kenaf fiber | - | - | - | 500–600 | 1.5–3.5 | 40–53 |
| 100% nano-PLLA | - | - | 135.6 | 55.6 | - | 3.3 |
| 90% nano-PLLA + 10% nano-HA | - | - | 142.5 | 53.2 | - | 3.5 |
| 80% nano-PLLA + 20% nano-HA | - | - | 156.8 | 48.6 | - | 3.8 |
| 70% nano-PLLA + 30% nano-HA | - | - | 130.3 | 42.3 | - | 3.9 |
| 60% nano-PLLA + 40% nano-HA | - | - | 125.9 | 38.6 | - | 4.1 |

## 3. Nano-Silica

In recent years, there has been an increasing use of silica nanoparticles, also called nano-silica or silicon dioxide nanoparticles, as a supplement to improve the physical and long-lasting qualities of concrete [51]. Research has demonstrated that including nano-silica into cement paste can enhance the concrete's durability by improving its nanostructure, as stated by [52]. Furthermore, Ref. [53] found that nano-silica can be a viable option to reduce cement consumption in the production of high-strength concrete (HSC), thereby enhancing cost-effectiveness and reducing the material's carbon footprint.

According to research by [54], nano-silica exhibits superior performance in terms of the filling effect and particle size distribution compared to conventional mineral admixtures. Incorporating nano-silica into concrete mixtures reduces porosity and enhances the pozzolanic breakdown of nano-silica with calcium hydroxide, causing the production of CSH and enhancing the mechanical properties. Additionally, research has shown that nano-silica can develop the cement setting process and enhance the cohesiveness of fresh mixes, as noted by [55].

Kumar et al. (2019) [56] have stated that nano-silica has high pozzolanic action, which accelerates the cement hydration at an early stage, leading to the conversion of calcium hydroxide into CSH gel, thereby improving the physical characteristics of concrete. Additionally, Figure 1a,b depicts, respectively, the morphology of nano-silica in powdered form and when seen via electron transmission microscopy.

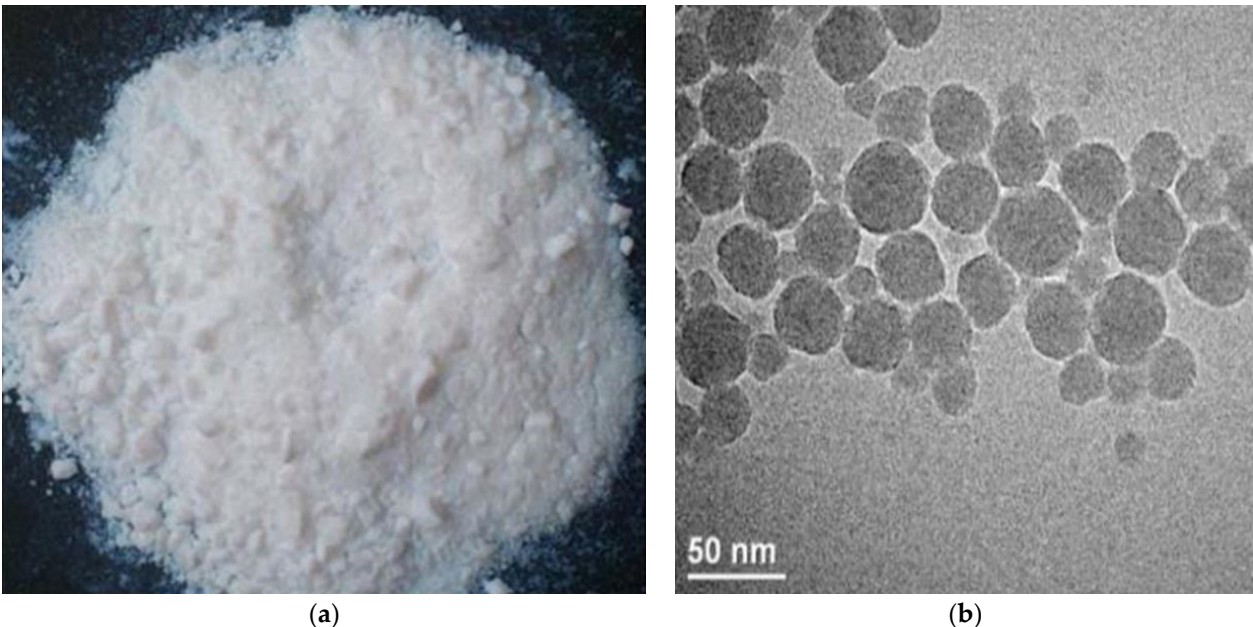

**Figure 1.** (**a**) The nano-silica powder [40], (**b**) transmission electron microscope image of nano-silica [40].

## 4. Mechanical Properties

*4.1. Compressive Strength*

Table 4 compiles the findings from multiple studies that have investigated the compressive strength of concrete. One such study by Chithra et al. (2016) [36] noted that the addition of 2 weight percent of nano-silica to concrete caused a 43% rise in compressive strength after 1 day and a 27% rise after 28 days, compared to standard concrete. However, the researchers cautioned against using excessive amounts of nano-silica, as it may lead to the aggregation of particles in the cement matrix, weakening the bonds within the matrix.

Isfahani et al. (2016) [42] investigated the effect of the water-to-binder ratio on mechanical strength enhancement and found a significant improvement as the w/c ratio increased. This improvement was attributed to the dispersion influence of nano-silica. Khaloo et al. (2016) [57] obtained similar results to those of [42]. Singh et al. (2016) [58] proposed that the ability of nano-silica particles to fill small voids in cement plays a crucial role in pozzolanic reactions.

Elkady et al. (2019) [59] studied how different nano-silica dosages affected the structural behavior of concrete. The findings revealed that using a 4.5% dosage of nano-silica caused a 13.5% rise in the compressive strength after seven days, compared to the standard group. For 1.5% and 3% nano-silica dosages, the strength gains were 3% and 4.5%, respectively. After 28 days, the strength gains were 17.5% and 29% at 1.5% and 4.5% nano-silica dosages, respectively, and a 43.5% increase in strength was observed at a 3% nano-silica dosage. According to [59], the study suggested that nano-silica particles agglomerated and prolonged the interaction time with the excess (CH), leading to the creation of CSH gel. These aggregated particles served as fillers, lowering porosity and boosting strength at an early age. In comparison to the control group, the optimal dose of nano-silica was determined to be 3%, which boosted the binding strength by 38.5%.

Yonggui et al. (2020) [60] studied the impact of different proportions of nano-silica replacements on the structural behavior of recycled aggregate concrete. Their research revealed that increasing the percentage of nano-silica replacements led to a decrease in compressive strength. They also found that higher temperatures during the production of recycled concrete, where nano-silica replaced cement, negatively affected the compressive strength. The temperature range between 25 and 200 °C caused the evaporation of both adsorbed and capillary water, resulting in gas pressure that weakened the concrete's interior

micro-structure. In contrast, Alhawat and Ashour (2020) [61] noted that adding 1.5% nano-silica to concrete instead of cement boosted bond strength and corrosion resistance.

Deb et al. (2015) [62] conducted research on the effect of nano-silica on geopolymer concrete and found that adding it increased the compressive strength by 2%, which matched to the control group. However, using more than 2% nano-silica led to the presence of unreactive particles, which weakened the concrete considerably. Adak et al. (2014) [63] stated that adding 6% nano-silica to geopolymer mortar improved its mechanical properties compared to normal cement mortar. Refs. [63,64] observed positive impacts on the physical characteristics of geopolymer concrete with the addition of 1% micro-silica and 2% nano-silica. Mustakim et al. (2021) [65] discovered that adding 1.5% nano-silica, in addition to silica fume, to geopolymer concrete improved the microstructure and resulted in outstanding strength under compression, presumably because of the quick alkali activation process of geopolymer concrete.

Jalal et al. (2015) [66] conducted a study demonstrating that the incorporation of 2% nano-silica in high-performance self-compacting concrete resulted in a substantial enhancement in its strength and durability when matched to the control group. Chithra et al. (2016) [36] similarly found that the substitution of cement with 2% colloidal nano-silica improved the structural behavior of HPC. Ghafari et al. (2014) [67] reported that using 3% nano-silica as a cement replacement in ultra-high-performance concrete resulted in an optimal performance by optimizing the pore structure and reducing the number of capillary holes, thereby improving the concrete's performance.

Multiple research studies have explored the potential benefits of incorporating nano-silica to enhance the structural behavior of HPC. Fallah and Nematzadeh (2017) [68] discovered that adding 2% nano-silica and 12% silica fume to cement improved the structural behavior of HPC. Similarly, Amin & Abu el-Hassan (2015) [33] used nano-silica, Cu, and Ni ferrite to create high-strength concrete with improved mechanical qualities. The increased amount of CSH gel that resulted from the nanoparticles' interaction might account for the higher strength seen in these trials.

According to [69], fiber-reinforced concrete with desirable mechanical properties can be achieved by adding 8% silica fume and 1% steel fibers, which prevent crack formation and enhance the material's performance. The study also found that substituting 2% of the cement with nano-silica further contributed to these benefits by promoting the production of more CSH gel and enhancing the concrete's strength.

**Table 4.** The impact of nano-silica on concrete compressive strength.

| Reference | % NS Content | Concrete Type | Remarks |
|---|---|---|---|
| Mukharjee & Barai, (2020) [70] | | Concrete | Studies have demonstrated that the compressive strength of mortar can be improved by increasing the amount of nano-silica, which improves the matrix. |
| Yonggui et al. (2020) [60] | 0, 3, and 6% | Recycled concrete | The compressive strength of concrete can be upgraded by adding nano-sillica, and studies have shown that higher percentages of nano-silica content can lead to an increase in the relative residual splitting tensile strength of concrete. |
| Their & Özakça (2018) [64] | 2% | Geopolymer concrete (GPC) | Unless paired with nano-silica, the addition of steel fiber did not result in a substantial improvement in compressive strength. |

**Table 4.** *Cont.*

| Reference | % NS Content | Concrete Type | Remarks |
|---|---|---|---|
| Nuaklong et al. (2018) [71] | 1, 2, and 3% | Recycled aggregate geopolymer concrete | GPC's mechanical and durability properties were both enhanced by 1% substitution of nano-silica. |
| Jalal et al. (2015) [66] | 2% | High-performance self-compacting concrete | The addition of 2% nano-sillica improved the performance of concrete. |
| Fallah & Nematzadeh (2017) [68] | 1, 2, and 3% | High-strength concrete | The adding of 2% and 12% of nano- and micro-silica improved the physical properties of concrete. |
| Hasan-Nattaj & Nematzadeh (2017) [69] | 1, 2, and 3% | Fiber-reinforced concrete | The 8% SF, 2% nano-silica, and 1% steel fibers in the concrete mix provide good mechanical properties. |
| Ganesh et al. (2016) [72] | 1 and 2% | High-strength concrete | The strength was increased when nano-silica was substituted at 2% to provide more strength. |
| Chithra et al. (2016) [36] | 0.5, 1, 1.5, 2, 2.5, and 3% | High-performance concrete | With more nano-silica content, the structural behavior improved, however a 2% replacement was determined to be ideal. |
| Atmaca et al. (2017) [73] | 3% | High-strength light weight concrete | In aggressive environments, nano-silica improves the concrete's strength |
| Supit & Shaikh (2015) [74] | 2 and 4% | High-volume fly ash concrete | The research suggests that adding 2% of nano-silica to cement can improve the performance of concrete. |
| Beigi et al. (2013) [75] | 0, 2, 4, 6% | Self-compacting concrete | Concrete's behavior may be improved by adding fibers and nano-silica. |

*4.2. Tensile Strength*

Adding nano-silica to concrete has been shown to increase its split tensile strength, according to research. Khaloo et al. (2016) [57] conducted an experiment on concrete using various sizes of nano-silica particles and discovered that 12 nm nano-silica was more effective at enhancing strength than 7 nm nano-silica. The study also suggested that the lower specific surface area of the 12 nm nano-silica facilitated better dispersion in water.

Fallah and Nematzadeh (2017) [68] examined the influence of the addition of nano-silica to concrete on its splitting tensile strength. Results indicated that substituting 3% of cement with nano-silica headed to a 16.10% rise in tensile strength compared to regular concrete. However, the reinforcing effect of silica fume was found to be stronger than that of nano-silica. Furthermore, when 4% of nano-silica was added to concrete when compared to unaltered concrete, it resulted in a 35% rise in splitting tensile strength.

According to Zhang et al. (2019) [76], the incorporation of nano-silica in concrete provides both a form of nano-reinforcement and fills the porosity in the concrete matrix. Figure 2 compares the tensile strength of concrete when 3% nano-silica is utilized with a 0.4 w/c ratio on various days.

The fact that nano-silica can both reinforce the concrete matrix and fill the porosity in the material is a promising characteristic for its use in concrete production. The comparison of tensile strength at different ages in Figure 2 provides evidence of the short-term durability of the nano-silica-modified concrete. Overall, the results indicate that adding nano-silica to

concrete may be a practical way to increase its split tensile strength, a crucial mechanical characteristic for withstanding tensile stresses and preventing material splitting.

The enhancement of the concrete's tensile strength is driven via multiple mechanisms. These encompass heightened bonding among nano-silica, cement, and aggregates; the filling of voids between cement particles and surrounding aggregates (a micro-filling effect that diminishes microcrack count and size); refining the interfacial transition zone (ITZ) by minimizing weak points and voids; the capacity for reducing the water-cement ratio due to the water-reducing effects; and a pozzolanic reaction between nano-silica and calcium hydroxide, leading to the creation of supplementary C-S-H gel. Therefore, self-healing concrete cracks and interfacial transition zones in concrete are considered the most important nanomaterial functionality [25,77,78].

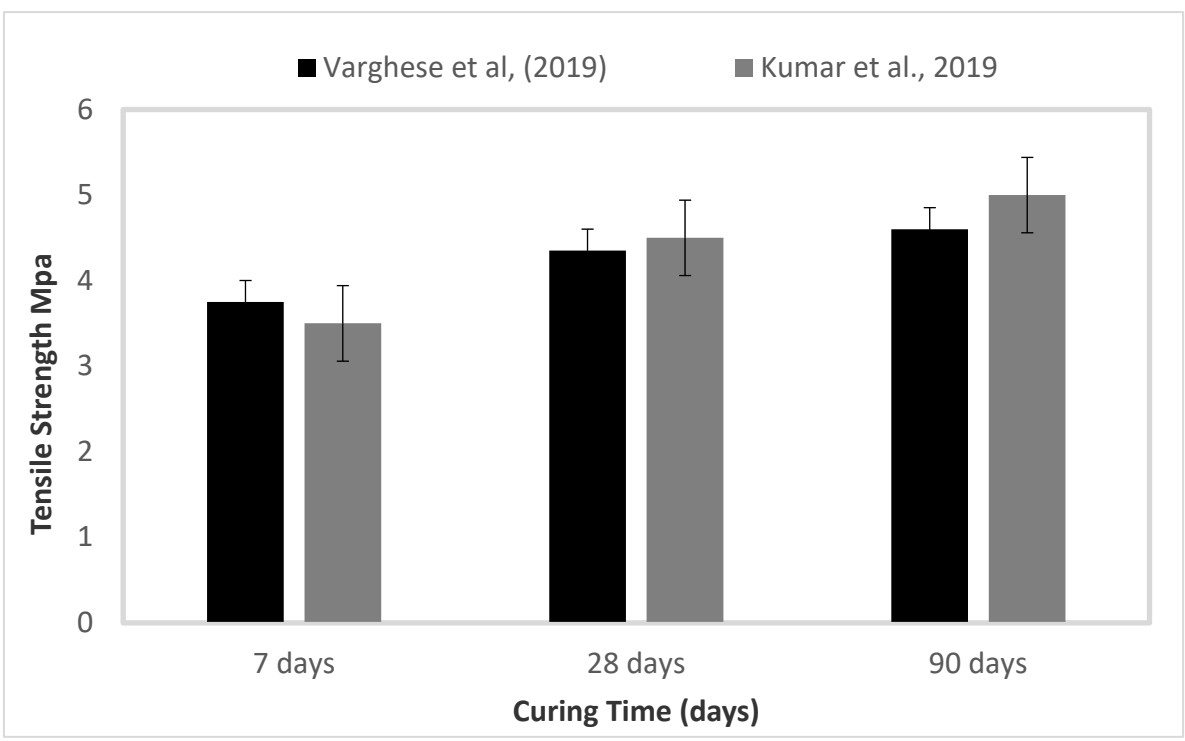

**Figure 2.** Concrete's tensile strength with 3% nS at 0.4 w/c ratio at 7, 28, and 90 days [56,79].

*4.3. Flexural Strength*

Guefrech et al. (2011) [80] found that the strength of flexural mortar was enhanced by increasing the nano-silica concentration from 3% to 10%. Similarly, (2015) [81] observed that a 3% nano-silica-modified mortar had the highest flexural strength after undergoing different curing periods.

Wu et al. (2019) [81] conducted research to investigate the mechanical behavior of nano-silica carbon fiber-reinforced concrete (NSCFRC) at various temperatures. The study revealed that the optimal concentration for improved flexural strength was 1 wt% nano-silica and 0.15 vol% carbon fiber at room temperature. Additionally, NSCFRC displayed an enhanced residual flexural strength at high temperatures in comparison to carbon fiber concrete. The findings suggest that incorporating nano-silica in carbon-fibered concrete can enhance its flexural properties even after being exposed to high temperatures. The chosen carbon fibers had a diameter of 7 μm and a length of 7 mm. Although the carbon fibers decrease the compressive strength of CFRC (Carbon Fiber-Reinforced Concrete), they contribute to an increase in the flexural strength of the concrete by reducing crack growth across the crack surface. On the other hand, the addition of nano-silica significantly enhances the mechanical properties of the concrete.

Abna and Mazloom (2022) [82] analyzed the impact of micro-silica, nano-silica, and polypropylene on the fractural strength of concrete. Their findings indicated that the addition of polypropylene fibers increased the fractural strength and fracture energy of the concrete samples. The optimal ratio of these components for achieving the extreme strength of the fracture and fractural energy was determined to be 5% micro-silica, 0.75% nano-silica, and 0.1% polypropylene. Furthermore, the study included Figure 3, which presents the fractural strength values of nano-silica-modified concrete.

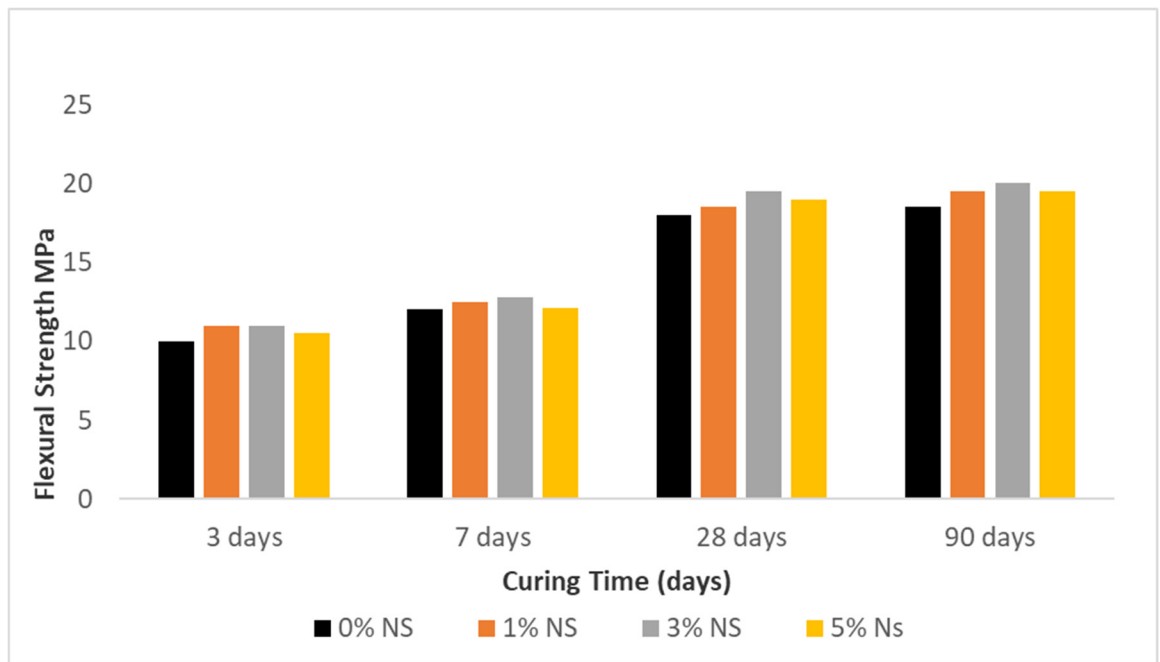

**Figure 3.** FS of nano-silica-modified concrete by [83].

Based on the studies, it can be inferred that the incorporation of nano-silica in concrete and mortar can enhance their flexural strength, especially when used in conjunction with other reinforcing constituents. Nevertheless, the ideal concentration of nano-silica may be influenced via different factors such as the water-cement ratio, the duration of curing, and the presence of other strengthening agents.

## 5. Durability

### 5.1. Chloride Penetration Resistance

Recent research has suggested that the addition of nano-silica to concrete can enhance its durability by reducing the penetration of chloride ions. For example, Ref. [84] observed that nano-silica, even at a low dosage of 0.3%, exhibited pozzolanic properties and filler activity, which led to a decrease in the permeation rate of water and ions of chlorine, as determined via MIP experiments. Similarly, Ref. [42] found that 0.5% of nano-silica could reduce the dispersion amount of chloride ions in concrete with water-to-binder ratios between 0.55 and 0.65, resulting in a more refined microstructure as well as a decrease in the crucial cutoff diameter of pores. By incorporating nano-silica (NS) into concrete, its durability can be enhanced via the reduction in pore size and connectivity. NS fills the gaps between cement particles, leading to a more compact microstructure. Additionally, NS exhibits high reactivity and undergoes pozzolanic reactions with calcium hydroxide, a byproduct of cement hydration. Comparing regular concrete to concrete with NS reveals significant distinctions in their MIP outcomes. The addition of NS induces changes in pore structure, size distribution, total porosity, pore volume, and distribution as well as capillary pore size, leading to a more refined microstructure characterized via reduced porosity and

enhanced durability. These studies consistently revealed a reduction in the charge passed in slag concrete, indicating a decrease in the transport of chloride ions.

Lincy et al. (2018) [85] observed that, compared to both micro-silica and the control samples, concrete modified with nano-silica demonstrated significantly greater resistance to chloride ion diffusion. Similarly, Jalal et al. (2015) [66] discovered that incorporating nano-silica and silica fume into (HPSCC) resulted in a reduction in the penetration of chloride ions. Figure 4, presented by Li et al. (2020) [86], shows the chloride penetration of autoclaved concrete at varying nano-silica content. As a result, adding nano-silica to concrete can increase its longevity by decreasing chloride ion diffusion; even a small addition of 0.3% can have a good effect.

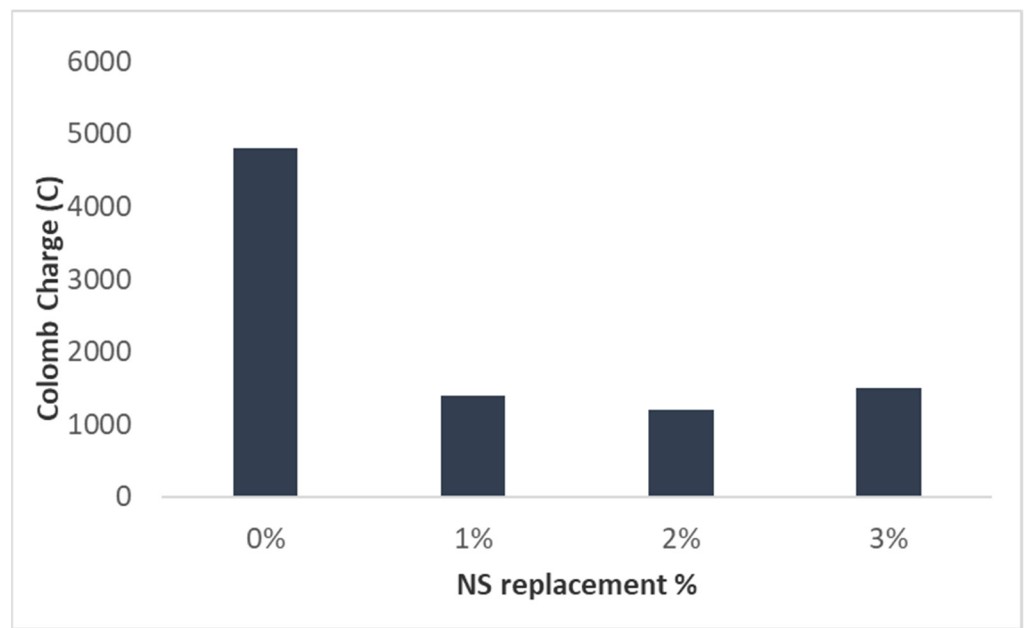

**Figure 4.** Chloride penetration of autoclaved concrete at different NS contents [87].

*5.2. Sulfate Resistance*

Huang et al. (2022) [87] examined the impact of adding 0–1.5 wt.% nano-silica and 0–1.0 vol.% polyvinyl alcohol (PVA) fibers on the sulfate resistance and mechanical properties of fly ash/cement paste hybrids. The study revealed that the addition of PVA and nano-silica fibers improved the mortars' physical characteristics and sulfate resistance. In comparison to the control group without these components, the hybrids with 1.0 vol.% PVA fibers and 1.5 wt.% nano-silica showed a 90% higher flexural strength after twenty eight days of curing. The compressive and flexural strengths of the cement mortars were significantly greater after 72 days of immersion in the $Na_2SO_4$ solution than they were after a total of 28 days of curing. Additionally, mortars containing 1.0–1.5 wt.% nano-silica had greater sulfate resistance after a hundred days in water, showing that the inclusion of nano-silica and PVA fibers can greatly increase resistance to sulfate assault.

Li et al. (2019) [88] discovered that combining micro-silica and nano-silica in concrete mixes enhanced the resistance to both sulfate and chloride attacks. The study demonstrated that the use of a combination of micro-silica and nano-silica was more effective in reducing the rate of strength and mass loss of the concrete samples compared to using either silica fume or nano-silica alone. Furthermore, the findings indicated that using micro-silica and nano-silica in concrete mixes resulted in the formation of more dense and compact microstructures, which can increase the strength of concrete to sulfate and chloride attacks.

Huang et al. (2020) [89] discovered that the combination of nano-silica in cement pastes improved their resistance to sulfate and that the level of improvement rose as the concentration of nano-silica enlarged. They also discovered that the addition of coarse nano-

silica was more active than fine nano-silica in improving sulfate resistance, presumably due to its superior filling capacity for voids in the cement matrix. Using a range of analytical methods, the study examined the microstructure and phase composition of the damaged specimens, providing a better understanding of the mechanisms underlying the improved sulfate resistance.

Previous studies have indicated that the inclusion of nano-silica in other cementitious materials, such as silica fumes, can improve the concrete's resistance to sulfate and chloride attacks. The addition of nano-silica is thought to refine the pore structure of the cement matrix, reduce pore connectivity, and enhance sulfate resistance. Furthermore, the efficacy of nano-silica in improving sulfate resistance may be influenced via its particle size, as coarse nano-silica particles have been found to be more effective than fine nano-silica particles in enhancing the sulfate resistance of cement mortars.

### 5.3. Water Absorption

Rajput & Pimplikar (2022) [90] conducted research which showed that increasing the concentration of nano-silica in M30 and M40 grade concretes decreased the absorption of water. The absorption of water from the M30 concrete reduced by 5.15%, 30.15%, and 35.66% compared to the control mix, as the concentration of nano-silica increased from 1% to 3%. For the M40 concrete, the water absorption decreased by 1.47%, 30.40%, and 58.97% when 1%, 2%, and 3% of nano-silica content was added, respectively. The addition of nano-silica to the cement composites improved the pore structure, resulting in reduced water absorption values and improved durability.

Athira and Shanmugapriya (2022) [91] investigated the potential of using calcined red mud cement pastes with and without colloidal nano-silica (CNS) at different (W/B ratios). They found that the incorporation of 1.5% CNS into red mud cement paste led to decreased water absorption at all W/B ratios, indicating that adding nano-silica can enhance the durability of cement-based materials.

According to [67], the inclusion of nano-silica in UHPC can decrease its water sorptivity and absorption. This effect is due to the high pozzolanic reactivity of nano-silica, which leads to the formation of more hydration products that fill the capillary pores and reduce their connectivity.

In general, the studies reviewed suggest that incorporating nano-silica into concrete can improve its pore structure, leading to lower water absorption and greater resistance to sulfate and acid attacks. Additionally, the pozzolanic reaction of nano-silica can decrease the connectivity of capillary pores by filling them with more hydration products, which can further enhance the concrete's durability by reducing water sorptivity and absorption.

### 5.4. Carbonation Resistance

According to several studies, adding nanomaterials may increase the concrete's resistance to carbonation. Li et al. (2017) [92] examined the influence of micro- and nano-silica on carbonation in concrete. The research found that adding both micro- and nano-silica to concrete can decrease carbonation, with the best results achieved when both additives were used together. The study also indicated that substituting 10% of micro-silica had a larger influence on reducing carbonation penetration than substituting 1% of nano-silica.

Kumar et al. (2019) [56] found that the addition of up to 3% micro-silica to regular concrete resulted in a reduction in carbonation depth by 46% and 17% after 7 and 70 days, respectively. However, increasing the amount of micro-silica beyond 3% resulted in an increase in carbonation depth over time. The study suggests that the combination of sufficient calcium hydroxide (CH) and 3% micro-silica can result in the development of C-S-H gel and a thicker matrix of concrete. However, adding more than 3% micro-silica did not lead to a denser concrete matrix.

Isfahani et al. (2016) [42] looked into the effects of various nano-silica doses on the carbonation resistance of concrete with various water-to-binder (w/b) ratios and discovered various outcomes. Contrary to other studies, the authors found that adding more nano-

silica did not increase the resistance of carbonation for concrete with 0.65 and 0.50 water-cement ratios. The study concluded that the w/b ratio is more important than nano-silica in improving the concrete's carbonation resistance, and adding more nano-silica could have an undesirable influence on the resistance of carbonation.

To note, the influence of nano-silica on carbonation resistance can differ based on the concrete mix and environmental conditions, as per previous studies. While some studies showed that adding nano-silica can enhance carbonation resistance, others observed little or negative effects. Factors like the dosage and type of nano-silica, w/c ratio, and curing conditions need to be considered when evaluating the possible impact of nano-silica on carbonation resistance. More research is necessary to establish the ideal conditions for using nano-silica to boost carbonation resistance in concrete.

## 6. Summary and Conclusions

In a review of around one hundred recent and past studies, it was found that increasing the amount of nano-silica in concrete enhances its compressive, split tensile, and flexural strength. This is due to the activator role of nano-silica in promoting hydration and improving microstructural pore density. Additionally, by raising the density of the interfacial transition zone (ITZ), nano-silica enhances the concrete matrix's resilience. However, due to issues such as agglomeration formation, high cost, and restricted availability in some areas, the use of nano-silica in the manufacturing of concrete is not commonly adopted. Another major worry is the ineffective distribution of nano-silica in concrete. While sonication is a viable remedy, further analysis is required to resolve this problem. Although nano-silica has been extensively researched, its commercial use in the construction industry is still in the early stages, and large-scale application remains limited:

1. By considering various factors such as the nature and dimensions of the nano-silica dosage, dispersion technique, dispersant type, water-cement ratio, and sequence of mixing, it becomes possible to discern the impact on the strength of concrete.
2. Inadequate dispersion or an increase in nucleation sites that can generate C-S-H gel due to the pozzolanic reaction can result in agglomeration when nano-silica has a high specific surface area. The dispersion method and type of dispersant used are factors that influence this.
3. The recommended replacement dose of nano-silica varies between 2 and 3%, according to the kind of cement used.
4. In order to retain the rolling effect of nano-silica and prevent the decrease in concrete workability, it might be essential to employ a significant quantity of plasticizers and elevate the water-cement ratio.
5. Nano-silica can enhance compressive strength while significantly improving other ductile properties, making it suitable to blend with fibers to further enhance ductility.
6. At the optimal dosage, the durability of nano-silica-modified concrete can be significantly improved. This is due to the stable hydration products generated in the pozzolanic process, which resist the ingress of harmful chemicals that cause degradation.

## 7. Future Research Directions

The following further works are suggested based on the evaluation:

1. It is important to assess the engineering characteristics of concrete with nano-silica added, such as bond, creep, shrinkage, etc.
2. Concrete with nano-silica added should have its fresh and hardened qualities evaluated to identify its thermal and acoustical characteristics.
3. A standardized mix design method for nano-silica-added concrete should be established to ensure consistency in the production process.
4. The optimal quantity of superplasticizers required for improved workability of nano-silica-added concrete needs to be determined.

5. The development of lightweight, highly durable, and nano-silica-infused concrete should be the main focus of research.
6. Thorough research is needed to optimize nano-silica-added concrete and create mathematical models that can predict concrete behavior accurately.

**Author Contributions:** Conceptualization, H.A. and F.A.; methodology, H.A.; investigation, T.E.; resources, F.A.; writing—original draft preparation, H.A and F.A.; writing—review and editing, T.E.; visualization, H.A.; supervision, F.A. and T.E. All authors have read and agreed to the published version of the manuscript.

**Funding:** This research received no external funding.

**Data Availability Statement:** Not applicable.

**Conflicts of Interest:** The authors declare no conflict of interest.

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
