# Peer review of "A Review of the Effect of Nano-Silica on the Mechanical and Durability Properties of Cementitious Composites"

_infrastructures, doi:10.3390/infrastructures8090132_

Round 1

Reviewer 1 Report

This paper presents a review on the effect of nano silica on the mechanical and durability properties of concrete. A significant issue is that it spends too much on the mechanical characteristics of raw nanomaterials, which is less relevant to the scope of the current study. This paper requires improvements before further consideration for publication. Some specific comments are listed below.

1. In the introduction section, there is one paragraph introducing that the nanomaterials can increases intelligence and multifunctionality, which is somehow not relevant to the topic of the study. The nanosilica is not electrically conductive and therefore cannot tailor the intelligence and multifunctionality for concrete. It is herein recommended to introduce more about the pozzolanic activity of the materials in terms of the mechanical and durability performances improvement rather than the smart properties of nanomaterials.

2. Line 38-39, the statement ‘The first focus was on modifying or improving the characteristics of cementitious materials utilizing oxides, both metallic and nonmetallic, at nanoscale’ seems hard to follow from the context.

3. The effects of nanomaterials on the functionality of concrete need more exploration and discussions, such as 10.1016/j.jclepro.2022.134116; 10.1016/j.conbuildmat.2022.127581; 10.1016/j.cemconcomp.2017.04.003.

4. Table 1 is recommended to supplement the quantitative results (any improvement in mechanical or durability properties) and the remarks of individual studies.

5. Line 98-99, for the statement ‘This is just one example of how nanoparticles can enhance the mechanical properties of materials.’, please avoid any of these meaningless sentences throughout.

6. Line 100-101, for the statement ‘While there is limited research on the physical characteristics of nanomaterials, significant strides have been made in understanding the mechanical characteristics of metal nanomaterials’, This statement is not reasonable since the physical properties of nanomaterials are usually provided by manufacturers.

7. Please add more comparative analysis and discussion on the compressive strength results in the literature.

8. Line 247-248, for the sentence ‘The comparison of tensile strength at different ages in Figure (3) provides evidence of the long-term durability of the nano-silica-modified concrete.’, the comparison of tensile strength cannot be correlated to the long term durability performance.

9. Line 401, what do you mean by ‘the influence on concrete strength may be detected’?

10.  More graphs are needed, putting together the results of various studies to produce new findings.

11.Some selected design mixtures of concrete presented in the previous studies should be added to the revised paper. They should be compared with each other in terms of strength, durability, etc.

Author Response

I wanted to take a moment to express my sincere gratitude for you invaluable contribution as the editor and reviwer for my current submission. Your dedication, expertise, and insightful feedback have greatly enriched the quality of my work, and I am truly appreciative of the time and effort you have invested in this process. Your meticulous review and constructive suggestions have undoubtedly played a pivotal role in refining the content and ensuring its accuracy. Your thoughtful insights have guided me in making meaningful improvements, enhancing the overall clarity and impact of the manuscript.
kindly, check the attached file. 

Reviewer 2 Report

The addition of nano silica can improve the mechanical properties and durability of concrete, and many other properties. Therefore, many researchers have studied and paid attention to the effect of nano silica in concrete. This manuscript summarizes the characteristics of nano silica and its impact on the mechanical properties and durability of concrete, and provides suggestions for future research directions. The main issues with the manuscript are as follows:

(1) The paper mainly lists the existing experimental results, lacking in-depth analysis of existing research results, and the theoretical depth of the manuscript is insufficient.

(2) Nano silica has a significant impact on the microstructure of concrete, but the manuscript did not delve into the impact of nano silica on the microstructure of concrete.

(3) The paper lacks original charts and the aesthetics of the charts are poor.

(4) The suggestions provided in the manuscript are not clear enough and lack guidance value for future research.

In summary, it is recommended to reject the manuscript.

Minor editing of English language required

Author Response

I wanted to take a moment to express my sincere gratitude for you invaluable contribution as the editor and reviwer for my current submission. Your dedication, expertise, and insightful feedback have greatly enriched the quality of my work, and I am truly appreciative of the time and effort you have invested in this process. Your meticulous review and constructive suggestions have undoubtedly played a pivotal role in refining the content and ensuring its accuracy. Your thoughtful insights have guided me in making meaningful improvements, enhancing the overall clarity and impact of the manuscript.
kindly, check the attached file 

Reviewer 3 Report

Dear authors,

The reviewed paper summarizes a literature review on the effect of nano-silica on the mechanical and durability properties of Concrete. The paper is well-written and follows a reasonable structure.

I have some comments that I would like the authors to address prior to the paper being published:

GENERAL COMMENTS

As a general comment, % of nano silica are given throughout the paper. Nonetheless, it is not specified if it is %wt. or % by volume or % by weight of cement or total binder, etc. This is a significant thing that should be addressed.

Table 2, column 2. Please use only one significant digit.

Tables 3 and 2 do not follow the MDPI format; please reformat them.

Lines 106-119. Reformat to MDPI guidelines.

The subindexes in the chemical formulations need to be written as subscripts and consistent throughout the manuscript. Please revise the manuscript.

Table 3.  Columns 4 and 5. Please, no significant digits.

Lines 139-141. Does nano-silica have a lower equivalent carbon footprint than cement? Nano silica, to the best of my knowledge, is produced by pyrolysis, which is an extremely energy-demanding process. Please provide references to support this statement.

Figures 1 and 2 should be merged into 1.a and 1.b.

What is the mechanism from a microstructural point of view under the improvement in tensile strength? Please expand on that in the manuscript.

The format of Figures 3-5 needs to be significantly improved. If the authors disclosed STDV, the error bars with the STDV should be shown in all the bar graphs.

Lines 259-265. What are the dimensions of the fibers used? How were the effects of the fibers vs. nano silica decoupled?

Figure 4. What are the yellow and gray bars? The legend does not show.

What is the mechanism behind the improvement of durability when using NS? How does the use of nano-silica affect the microstructure? Could you show differences in mercury intrusion porosimetry (MIP) between the two? (there is a body of knowledge showing that).

Can you show (with author permission or adapting the figure and citing) SEM comparisons (pictures) of regular concrete vs. NS modified concrete? This would help a lot in reinforcing the explanations summarized in the text, especially while discussing the increase in density of the ITZ.

SPECIFIC COMMENTS:

Please support lines 33-34 with references.

Rephrase lines 61-63 for clarity.

There are some minor spelling errors throughout the text. Please revise the manuscript.

The word nano-sic is used throughout the text. Please rephrase to nano-silica.

Replace "reduction" by "reduce" in line 287.

Author Response

(The authors gave the same response as above.)

Reviewer 4 Report

This article is a review of the effects of nano-silica (NS) on the mechanical and durability properties of concrete. It discusses how NS has been investigated as a partial cement alternative in concrete and how it can increase both strength and durability while reducing structural micropores. The review also proposes areas for future research.

The article has some noticeable shortcomings and disadvantages that could be improved. Here are some examples:

1.       The title is too broad and vague. It does not specify what kind of nano silica, concrete, or properties are being reviewed. A more specific and informative title could be “A review of the effect of nano silica on the mechanical and durability properties of cementitious composites”.

2.       The abstract does not clearly state the main objectives, findings, and implications of the review. A good abstract should be concise, accurate, and informative, and should highlight the most relevant aspects of the review. It should also include keywords that reflect the content and scope of the review.

3.       The introduction is not well organized and does not provide a clear background and motivation for the review. It does not explain why nano silica is an important nanomaterial for concrete applications, what are the main challenges and gaps in the existing literature, and what are the specific aims and scope of the review. A good introduction should provide a clear context and rationale for the review, as well as a brief overview of the main topics and structure of the review.

4.       The literature review is not comprehensive and systematic. It does not cover all the relevant studies and sources on the topic, and does not critically evaluate and compare their methods, results, and limitations. It also does not follow a clear structure or logic, and jumps from one topic to another without clear transitions or connections. A good literature review should provide a thorough and coherent synthesis of the existing knowledge on the topic, as well as identify and discuss the main controversies, inconsistencies, and gaps in the literature.

5.       The discussion and conclusion are not well developed and do not provide a clear summary and synthesis of the main findings and implications of the review. They also do not address the limitations of the review, suggest directions for future research, or provide practical recommendations for concrete applications. A good discussion and conclusion should highlight the main contributions and implications of the review, as well as acknowledge its limitations and suggest areas for further investigation.

To revise the article, some possible strategies are:

1.       Rewrite the title to make it more specific and informative.

2.       Rewrite the abstract to make it shorter and clearer, and include keywords.

3.       Rewrite the introduction to provide a clear background and motivation for the review, as well as a brief overview of the main topics and structure of the review.

4.       Rewrite the literature review to make it more comprehensive and systematic, and follow a clear structure or logic that covers all the relevant aspects of nano silica effects on concrete performance.

5.       Rewrite the discussion and conclusion to provide a clear summary and synthesis of the main findings and implications of the review, as well as address its limitations and suggest directions for future research.

I hope this helps you with your task.

Author Response

I wanted to take a moment to express my sincere gratitude for you invaluable contribution as the editor and reviwer for my current submission. Your dedication, expertise, and insightful feedback have greatly enriched the quality of my work, and I am truly appreciative of the time and effort you have invested in this process. Your meticulous review and constructive suggestions have undoubtedly played a pivotal role in refining the content and ensuring its accuracy. Your thoughtful insights have guided me in making meaningful improvements, enhancing the overall clarity and impact of the manuscript.
kindly, check the attched file

Round 2

Reviewer 1 Report

Response to Q3, the authors stated "Self-healing concrete cracks and interfacial transition zone in concrete are considered the most important nanomaterial functionality." I don't see any reference to support it. Please add relevant reference and cited.

Author Response

Three more references have been added and highlighted in green. 

Reviewer 2 Report

The main issue has been modified.

Author Response

Thank you!

Reviewer 3 Report

The authors have made a significant effort to improve the manuscript. Therefore I suggest the manuscript is accepted as is.

Author Response

Thank you!

Reviewer 4 Report

The authors have responded most of the points raised in the comments.

Minor editing of English language required.

Author Response

Thank you!

Round 3

Reviewer 1 Report

The paper can be accepted.